# Long-Term Impact of Suppressive Antibiotic Therapy on Intestinal Microbiota

**DOI:** 10.3390/genes12010041

**Published:** 2020-12-30

**Authors:** Rosa Escudero-Sánchez, Manuel Ponce-Alonso, Hugo Barragán-Prada, María Isabel Morosini, Rafael Cantón, Javier Cobo, Rosa del Campo

**Affiliations:** 1Servicio de Enfermedades Infecciosas, Hospital Universitario Ramón y Cajal, Instituto Ramón y Cajal de Investigación Sanitaria (IRYCIS), and Red Española de Investigación en Patología Infecciosa (REIPI), 28034 Madrid, Spain; rosa.escudero@salud.madrid.org (R.E.-S.); javier.cobo@salud.madrid.org (J.C.); 2Servicio de Microbiología, Hospital Universitario Ramón y Cajal, Instituto Ramón y Cajal de Investigación Sanitaria (IRYCIS), and Red Española de Investigación en Patología Infecciosa (REIPI), 28034 Madrid, Spain; lugonauta@gmail.com (M.P.-A.); mariaisabel.morosini@salud.madrid.org (M.I.M.); rafael.canton@salud.madrid.org (R.C.); 3Servicio de Microbiología, Hospital Universitario Ramón y Cajal, Instituto Ramón y Cajal de Investigación Sanitaria (IRYCIS), 28034 Madrid, Spain; hugobprada@gmail.com

**Keywords:** suppressive antibiotic therapy, gut microbiota, PCR-DGGE, bacterial viability, propidium monoazide, antibiotic multirresistant colonization

## Abstract

The aim was to describe the safety of indefinite administration of antibiotics, the so-called suppressive antibiotic therapy (SAT) and to provide insight into their impact on gut microbiota. 17 patients with SAT were recruited, providing a fecal sample. Bacterial composition was determined by 16S rDNA massive sequencing, and their viability was explored by PCR-DGGE with and without propidium monoazide. Presence of antibiotic multirresistant bacteria was explored through the culture of feces in selective media. High intra-individual variability in the genera distribution regardless of the antibiotic or antibiotic administration ingestion period, with few statistically significant differences detected by Bray-Curtis distance-based principle component analysis, permutational multivariate analysis of variance and linear discriminant analysis effect size analysis. However, the microbiota composition of patients treated with both beta-lactams and sulfonamides clustered by a heat map. Curiously, the detection of antibiotic resistant bacteria was almost anecdotic and CTX-M-15-producing *E. coli* were detected in two subjects. Our work demonstrates the overall clinical safety of SAT and the low rate of the selection of multidrug-resistant bacteria triggered by this therapy. We also describe the composition of intestinal microbiota under the indefinite use of antibiotics for the first time.

## 1. Introduction

Increased life expectancy with an acceptable quality of life is a relevant characteristic of current society. The rate of joint replacement surgery is increasing worldwide, with estimates of up to 80,000 operations annually in Spain and an incidence of further prosthetic joint infection of 3–4% [1]. Prosthetic infection increases not only morbidity and mortality rates (approximately 2–7% for patients over 80 years of age) but also healthcare costs [2]. The factors that contribute to implant associated infections are related to the implanted biomaterial, surgical procedure, soft tissue and immunological system status, and infective microorganism-related factors [2]. Managing this situation, including the surgical and nonsurgical approaches, is complex. A curative approach is not possible for a small proportion of patients, whose only option for controlling symptoms and maintaining functionality is the indefinite administration of antibiotics, also know as suppressive antibiotic therapy (SAT), an apparently safe and well-tolerated treatment [3,4,5,6]. The antimicrobial impact of SAT is not limited to the infection site; one of the most concerning adverse effect is the disturbance of the commensal gut microbiota. Although several studies have assessed the long-term and short-term effects of particular antibiotics on gut microbiota [7,8,9,10,11], this feature has not been explored in patients undergoing SAT, who are usually elderly and have numerous comorbidities and are therefore candidates for colonization/infection by pathogens such as *Clostridioides difficile* and multidrug-resistant bacteria. The aim of this study was to describe the safety of SAT over extended periods and its ecological impact on gut microbiota. 

## 2. Materials and Methods 

### 2.1. Study Design

We conducted a prospective observational study at our center’s Department of Infectious Diseases from 2016 to 2018 on patients undergoing SAT due to unresolved implant-associated infection managed with implant maintenance. The inclusion criteria were an age over 18 years, regular follow-up in our consultation, signing the written informed consent document and undergoing SAT for at least an entire month. The exclusion criteria were a history of inflammatory bowel disease (IBD), colorectal cancer, *C. difficile* infection in the previous 3 months and the consumption of other antibiotics due to a different infectious episode in the previous 30 days. Each patient provided a fecal sample, and their demographic and clinical characteristics were recorded from their clinical charts in an anonymous confidential database. This study was approved by the center’s local ethics committee (27 March 2017 acta 321).

### 2.2. Sample Processing

After collecting the feces, the samples were divided into two aliquots and immediately frozen at −80 °C. One aliquot was employed to determine the colonization of viable antibiotic multidrug-resistant bacteria by conventional microbiological cultures, whereas the remaining aliquot was reserved to determine the whole bacterial composition by 16S rDNA amplification and massive sequencing. 

To select multidrug-resistant bacteria, the samples were cultivated in M-*Enterococcus* agar plates supplemented with 6 mg/L of vancomycin, mannitol-salt agar plates supplemented with 4 mg/L of oxacillin, MacConkey agar plates supplemented with 1 mg/L of cefotaxime and with 2 mg/L of imipenem. We performed extended incubation at 37 °C for at least 5 days to allow for the growth of fastidious organisms. Colonies were identified by matrix-assisted laser desorption/ionization-time of flight Mass Spectrometry (Bruker Daltonics, Leipzig, Germany), and their antibiotic susceptibility was determined using the MicroScan WalkAway system (Beckman Coulter, Brea CA, USA). 

To determine the whole microbiota composition, we completely solubilized portions of 0.5 mg of feces in 5 mL of sterile water and employed 0.5 mL of this solution to obtain total DNA using the QIAamp kit (Qiagen, Hilden, Germany), determining their concentration and quality using the Qubit fluorometer (Thermo Fisher Scientific, Waltham, MA, USA). We amplified the V3-4 region of the bacterial 16s rDNA gene by PCR and then used the amplicons to prepare the sequencing library, following the Illumina 16S Metagenomic Sequencing Library Preparation Guide. Lastly, 16S rDNA massive sequencing was performed on a MiSeq platform (Illumina, San Diego CA, USA) at the Foundation for the Promotion of Health and Biomedical Research (FISABIO) in Valencia, Spain. 

### 2.3. Bioinformatics and Statistical Analysis

The 16S rDNA sequencing data analysis and statistics were performed using the QIIME2 (version 2018.4) software package [12]. Taxonomic annotation of the operational taxonomic units was performed by employing the classify-sklearn naïve Bayes taxonomy classifier via the “q2-feature-classifier” plugin for QIIME2, using the SILVA 132 database as reference [13]. The relative abundance and contingency tables included singletons and very low represented taxons. Alpha and beta diversity parameters were calculated using the “diversity” plugin for QIIME2. The taxonomic alpha diversity and community richness were estimated using the Shannon and Chao1 indices, respectively. Significant differences in these parameters, according to the type of antibiotic employed for treatment, were tested using a Kruskal-Wallis pairwise analysis. Values were considered statistically significant if <0.05. The bacterial composition between samples (or beta diversity) was estimated using the Bray-Curtis distance, which determines a matrix of dissimilarity scores based on the taxa present and their relative abundance [14]. After grouping the samples by antibiotic therapy type, a permutational multivariate analysis of variance was applied to test whether the intragroup distances differed significantly from the intergroup distances. We also graphically explored the Bray-Curtis distances by a principal coordinate analysis using the “emperor” plugin for QIIME2 [15]. We built a heat map using the 50 genera with a higher median relative abundance using the “heatmap” plugin for QIIME2. Lastly, we performed a linear discriminate analysis effect size (LEfSe) to assess the differential abundance of selective taxa between the antibiotic therapy groups [16]. The alpha value for the factorial Kruskal-Wallis test was 0.05, and the threshold for the logarithmic linear discriminant analysis score for discriminative features was set at 2.0. All sequencing reads that map to the human reference genome have been removed from the sequencing files.

### 2.4. Bacterial Viability

As bacterial viability cannot be inferred from the high-throughput sequencing data, we aimed to evaluate the SAT impact on alive bacteria using propidium monoazide (PMA), a photoreactive DNA dye that does not interact with intact biological membranes, instead targeting free DNA and dead cells. Once PMA is intercalated into a DNA molecule, it prevents its amplification by PCR, obtaining only positive amplification from viable bacteria. 

Consequently, all fecal samples were processed in parallel with and without PMA in addition to a control group of dead bacteria after boiling for 15 min. The PMA was subsequently activated by incubating in the dark for 5 min and photoactivating for 30 min with blue LED light. We then extracted total DNA using the QIAamp kit (Qiagen, Hilden, Germany) and performed universal bacterial amplification using rDNA 16S primers. Amplicons were further separated and visualized on a gradient of denaturing acrylamide gel (DGGE) to differentiate live and dead bacterial species.

## 3. Results

### 3.1. Clinical Data and Adverse Effects

Of the 20 patients treated in our department who met the inclusion criteria, 17 of them signed the informed consent and provided a fecal sample. Table 1 shows the patients’ demographic and clinical data, with a homogeneous sex distribution (8 female and 9 male patients) and a mean age of 75.9 ± 12.5 years. One of the patients (34 years old) was considerably younger than the others, who were between 66 and 88 years old. The Charlson index that predicts 10-year survival in patients with multiple comorbidities had a median value of 1.24 ± 1.48/IQR 1 (0–1.5), estimating a probability of 10-years survival of 96%. The unresolved prosthetic infections were mainly caused by *Staphylococcus aureus* (8 patients, 47.1%), and *Staphylococcus epidermidis* (3 patients, 17.6%). By the sample collection time, the patients had taken antibiotics for a median of 32 months (range, 1–109 months).

Each antibiotic therapy was individually considered based on the patients’ characteristics, tolerability, safety profile, drug-drug interactions, and the antibiotic susceptibility of the infective microorganism. The prescribed oral antibiotics were beta-lactams (6 patients, 35.2%), cotrimoxazole (4 patients, 23.5%), fluoroquinolones (4 patients, 23.5%, one of them in combination with rifamycin), tetracyclines (2 patients 11.7%), and clindamycin (1 patient 5.8%). The remaining patient was hospital-admitted with cancer, had a central catheter and was administered intravenous teicoplanin. 

The patients were questioned as to SAT-related adverse effects, with 5 (29.4%) reporting mild gastrointestinal symptoms. Three patients had asymptomatic bacteriuria, whereas a fourth patient experienced a single episode of urinary tract infection caused by a microorganism resistant to the antibiotic used in the SAT regimen (doxycycline); however, intestinal colonization by the infective microorganism was not demonstrated. Lastly, one of the female patients reported vaginal candidiasis and weight gain, in clear association with amoxicillin intake (Table 1).

### 3.2. Fecal Colonization by Antimicrobial Resistant Bacteria

The fecal carriage of antibiotic multidrug-resistant microorganisms was not detected by selective culturomics, with the exception of two extended-spectrum beta-lactamase (ESBL) producing *Escherichia coli* isolates in patients 7 and 15, which were further characterized as CTX-M-15. Both patients were treated with non-beta-lactams antibiotics: levofloxacin/rifamycin and teicoplanin, respectively.

### 3.3. Gut Microbiota Composition

For the gut microbiota characterization, we evaluated the richness and diversity of the gut microbiota using the Shannon and Chao1 alpha diversity indices (Figure 1), detecting a similar pattern for beta-lactam and sulfonamide-treated patients, whereas tetracyclines (and particularly fluoroquinolones) provoke a considerable decrease in the Chao1 index. The most aberrant ecosystem corresponded to patient 7, who had been taking levofloxacin plus rifamycin for a single month. On the contrary, patient 15 was also treated with intravenous teicoplanin during a month but he presented high alpha-diversity values. 

Figure 2 shows the phyla distribution according to the massive sequencing analysis. There was high variability among all patients and phyla, particularly for *Firmicutes* (median value, 76.6%; range, 45–95%), *Bacteroidetes* (8.7%, 0.7–45%), *Proteobacteria* (2.1%, 0.006–28%), *Actinobacteria* (3.1%, 0.2–22%), and *Verrucomicrobia* (0.01%, 0–15%). The abundance of the referent genera from the each phyla was compared by the prescribed antibiotic and the intake duration: *Faecalibacterium* from *Firmicutes, Bacteroides* from *Bacteroidetes, Escherichia/Shigella* from *Proteobacteria, Akkermansia* from *Verrucomicrobia,* and *Bifidobacterium* from *Actinobacteria* (Figure 3). As expected, the analysis showed high intraindividual variability; however, *Akkermansia* was more depleted by sulfonamides, whereas quinolones appeared to negatively affect *Bifidobacterium* abundance. A low abundance of *Escherichia/Shigella* was observed in all patients, except for patient 10.

The beta diversity analysis using the Bray-Curtis distance-based principle component analysis and the permutational multivariate analysis of variance detected no clustering patterns according to antibiotic class or SAT duration (Figure 4). There were also no significant differences in the LEfSe analysis using default settings; however, to identify the microorganisms affected by each antibiotic, we performed a less strict comparison (Figure 5). The analysis revealed that beta-lactams inhibit *Escherichia/Shigella*, *Dorea* and *Solobacterium* genera, the *Acidaminococcaceae* family and the *Actinomyces* phyla while favoring the expansion of *Flavonifractor, Sutterella, Christensenella, Anaerofustis, Haemophilus* and *Gelria* genera. The impact of sulfonamides was considerably lower, only inhibiting the *IncertaeSedis* and favoring *Faecalibacterium, Prevotella, Pseudobutyrivibrio, Dorea, Collinsella, Mogibacterium, Alloprevotella, Olsenella and Howardella*.

Despite each individual having their particular microbiota and the fact that the antibiotic effect might not be generalizable to all individuals (mostly due to the differing gut resistome of each participant), we decided to interconnect the bacterial abundances with the SAT antibiotic using a heat map analysis (Figure 6). The results demonstrated that 3 out of 6 patients receiving beta-lactam clustered together, whereas in the 4 patients receiving sulfonamides the grouping occurred in pairs.

### 3.4. Bacterial Viability

Lastly, we assessed the viability of the fecal bacteria using PCR-denaturing gradient gel electrophoresis (with and without PMA) to inhibit the amplification of non-viable cells, thereby avoiding unspecific DNA amplifications of transient bacteria (Figure 7). We observed a similar intensity of bands in the experiments with and without PMA, which makes us presuppose that most of the intestinal bacteria preserved their viability, without significant differences between patients.

## 4. Discussion

The short and long-term impact of antibiotics on intestinal microbiota has been documented by nonculture-based techniques [17,18,19,20,21,22], demonstrating high intraindividual and interindividual variability, even for similar antimicrobial compounds. Most of those studies indicated an individual antibiotic response strongly influenced by the gut ecosystem’s initial diversity and strength [23]. In the present study, we explored for the first time (to our knowledge) the impact of SAT on the gut microbiota of an aging population (all less one) with comorbidities (median Charlson index of 1.24) and unresolved bacterial infections. Our expected results included fecal colonization by antibiotic multidrug-resistant bacteria, not only by continuous antibiotic selection but also as a consequence of the frequent hospital visits of these elderly patients, as well as the disruption of the gut ecosystem. Surprisingly, neither of these two expectations came true.

SAT is not just a therapeutic option but is in fact the last option for treatment for implants-associated infections when the curative surgical approach is not feasible. SAT safety was our first goal, and we observed no serious complications in our cohort except for patient 11, who reported vaginal candidiasis and weight gain after amoxicillin intake. One of the most relevant consequences of prolonged antibiotic therapy is the infection by toxigenic *C. difficile* due to disruption of the commensal microbiota. The protective role of *Barnesiella* genera and the *Lachnospiraceae* family has been reported [24], and both were abundant in our cohort (0.7 ± 1.8%; range, 0–6.9% for *Barnesiella*; and 19.3 ± 15%; range 3.6–72.0% for the *Lachnospiraceae* family), and that could be the reason of the low *C. difficile* prevalence among our patients.

The weight gain and vaginal candidiasis reported by patient 11 (who had taken amoxicillin for 11 months) are related to major changes in gut microbiota. The growth-promoter property of antibiotics is well known because the property has been extensively employed for livestock but at low doses. Although other patients in our study had been treated with beta-lactams, patient 11 was unique in taking amoxicillin, and we cannot rule out that an increase in body mass index was related to the SAT intake, as has been previously published by other authors [25,26,27]. Lastly, the mild intestinal adverse effects such as diarrhea and flatulence observed in our study tended to disappear during the follow-up or were controlled.

Unexpectedly, antibiotic multidrug-resistant microorganisms were not detected in feces, except for two ESBL producing *E. coli* isolates but in two patients who had not been treated with a beta-lactam. Despite the oral intake, most of the antibiotic dose is absorbed in the small intestine, passing to the blood compartment, with only small amounts of the antibiotic reaching the colon. However, non-modified antibiotic molecules could also be reintroduced into the lumen through biliary excretion, particularly the lipophilic antibiotics such as tetracyclines and fluoroquinolones. Available data on the selection process of antibiotic-resistant bacteria are inferred from in vitro experiments, and some discrepancies can occur in vivo [24]. One factor to consider is that antimicrobials (such as azithromycin) can exert adverse effects regardless of their antimicrobial activity, as well as potent anti-inflammatory effects [28]. The interplay between microbiota and immunity is one of the main factors influencing homeostasis [29,30]; although the potential antibacterial activity of other drugs or foods should also be considered [31].

Despite previous efforts, specific intestinal microbiota targets have not yet been established for each family of antimicrobials. Aminoglycosides appear to select *Bacteroidetes* and *Firmicutes* phyla, whereas quinolones activity largely varies in different individuals [32]. Fluoroquinolones appear to have a deeper impact [20,22], although other authors have reported a low effect for fluoroquinolones and vancomycin (intravenous), whereas trimethoprim-sulfamethoxazole had very little impact [33]. The addition of a carbon-derived supplement that absorbs excess antibiotic in the lumen has been proven to protect intestinal microbiota [34], and this strategy should be considered for extended antibiotic therapies such as SAT. All available data indicate individualized activity of the antibiotics in the gut microbiota according to the previous individual composition of the microbiota [20,35]. Our population was characterized by their advanced age and clinical status, the median value of Charlson index was 1.24 ± 1.48/IQR 1 (0–1.5). Despite the lack of a baseline sample for all patients prior to starting the SAT, we observed a remarkable conservation of the bacterial alpha-diversity and beta-diversity after the extended antibiotic exposure. Moreover, we detected certain patterns, mainly in the beta-lactams and sulfonamide, despite those being the most widely used antibiotics.

The long-term antibiotic effects on the entire gut ecosystem have not been extensively studied, and bacterial signaling is probably different when they are excited/stimulated by antimicrobial molecules. *Bacteroidetes, Bifidobacterium,* strict anaerobes, and minority populations seem to be the most fragile bacterial populations [18,36,37,38,39], and our results showed the disparity in the abundance of some of these populations.

Our study’s main limitations are the small number of patients for each antimicrobial type, the specific clinical features of each individual that prevent an homogeneous comparison and the lack of baseline samples to better determine the long-term impact of the antibiotics. In contrast, the study’s strength was the inclusion of the PMA technique to assure bacterial viability, which was demonstrated as high and comparable in all patients.

## 5. Conclusions

In summary, our study demonstrates the overall safety of SAT and the low-rate of antibiotic multi-drug resistant bacterial selection triggered by this therapy. We also described the composition of the intestinal microbiota under the indefinite use of antibiotics for the first time. Our data suggest that SAT is safe in a clinical context of specialized clinical follow-up that monitors adverse effects, interactions and toxicity, introducing the necessary modifications in the treatment.

## Figures and Tables

**Figure 1 genes-12-00041-f001:**
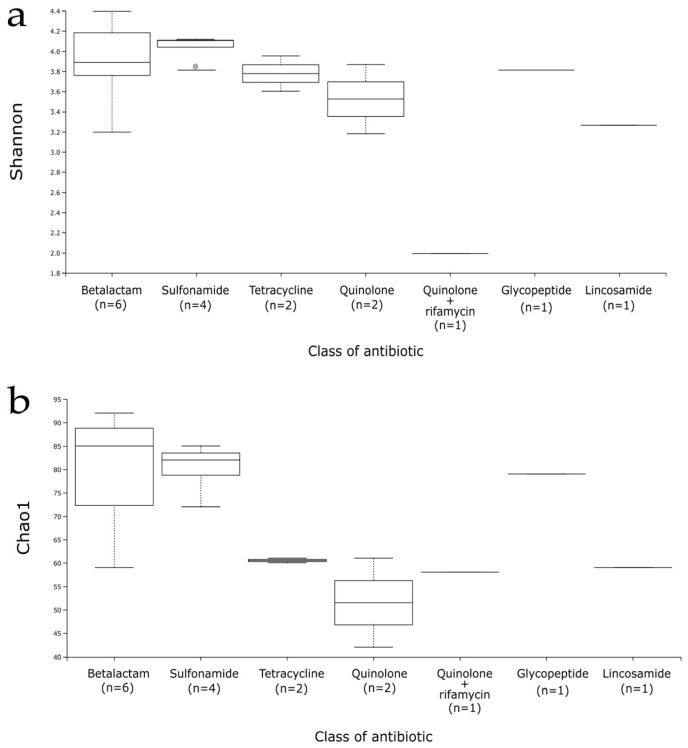
Alpha diversity boxplot of samples according to the antibiotic class. (**a**) Shannon index, (**b**) Chao1 index.

**Figure 2 genes-12-00041-f002:**
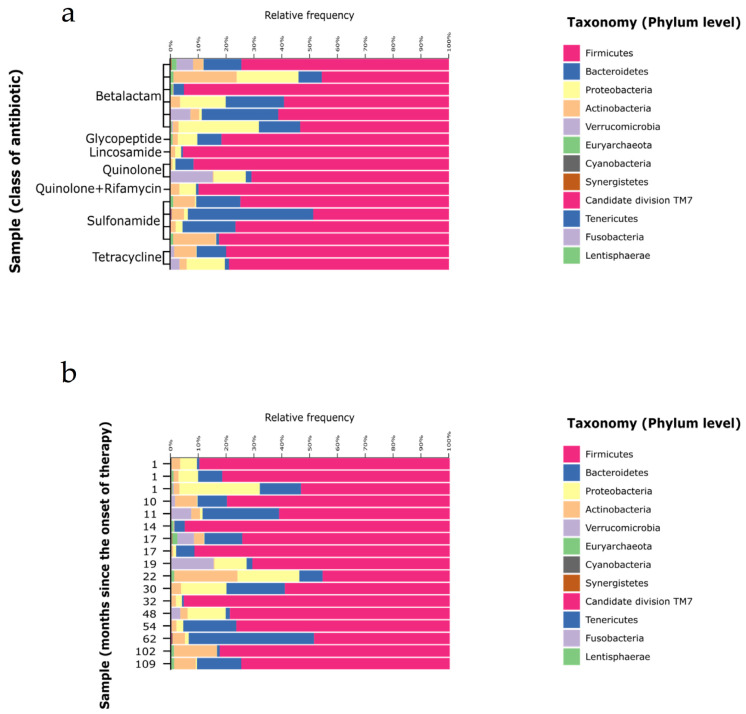
General overview of bacterial phyla distribution of the 17 samples considering the antibiotic class (**a**) and the length of the treatment (**b**).

**Figure 3 genes-12-00041-f003:**
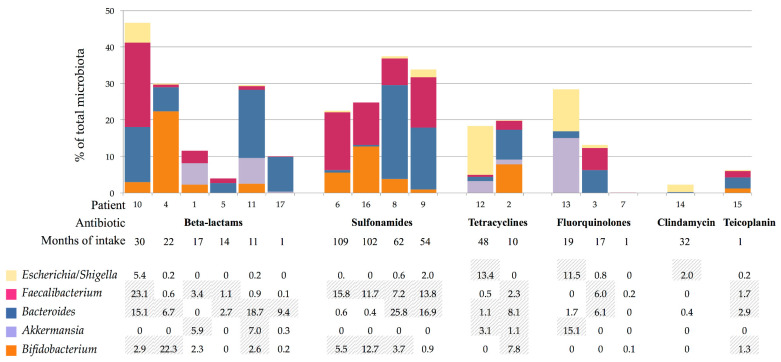
Abundance percentages of particular genera of majoritarian phyla for each patient: in columns is represented the addition of each genera proportion respect to the total microbiota percentages, above detailed. Striped cells correspond to those with higher abundance.

**Figure 4 genes-12-00041-f004:**
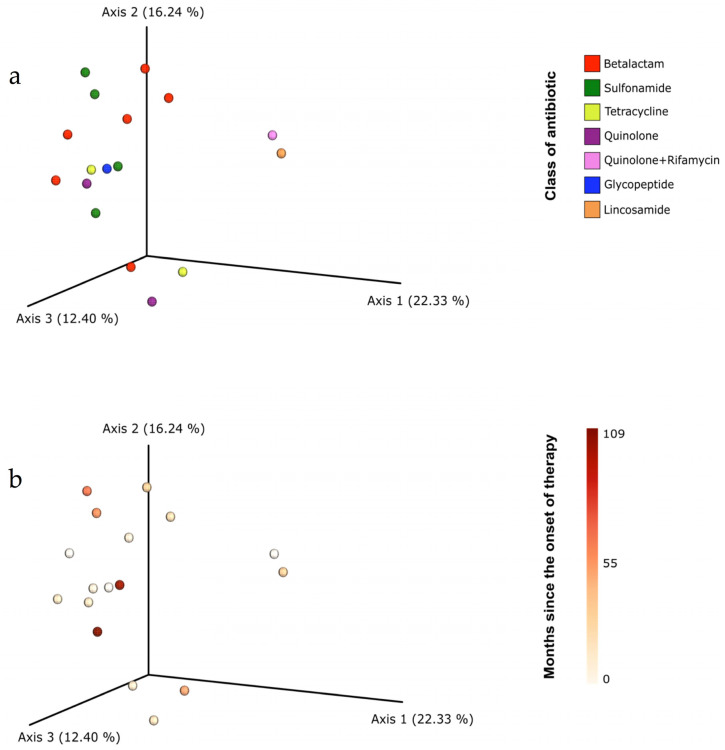
Principal coordinates analysis (PCoA) plot displaying the beta-diversity spatial ordination among samples taking account the antibiotic class (**a**) and the period of intake (**b**).

**Figure 5 genes-12-00041-f005:**
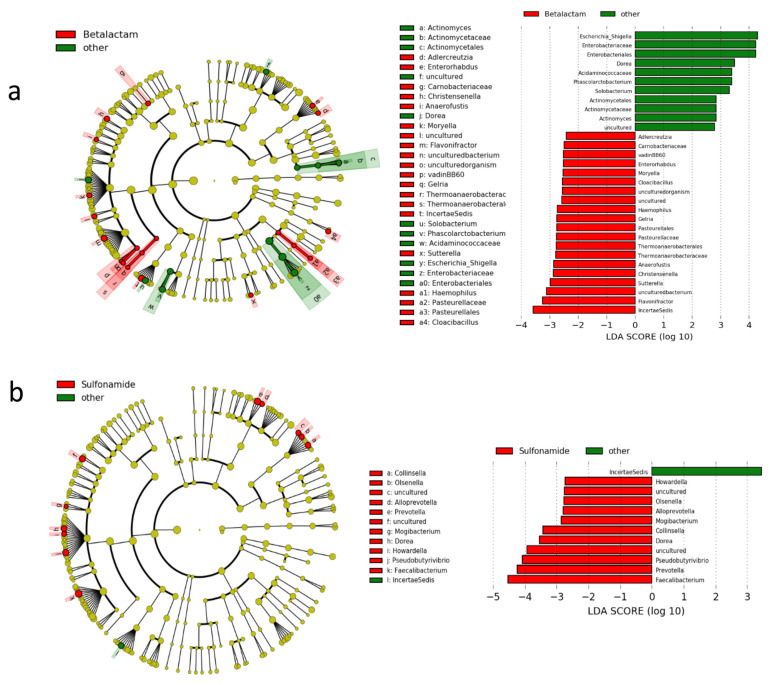
Differentially abundant microbial clades according to linear discriminate analysis effect size (LEfSe). The left side of each graph depicts a cladogram, where dark-yellow circles represent non-significant microbial clades, green circles represent significantly more abundant clades and red circles represent significantly less abundant clades. On the other side of each graph, significant clades and their associated LDA score are shown. (**a**) Comparison between patients receiving betalactam antibiotics and the other patients (**b**) Comparison between patients receiving sulfonamides and the other patients.

**Figure 6 genes-12-00041-f006:**
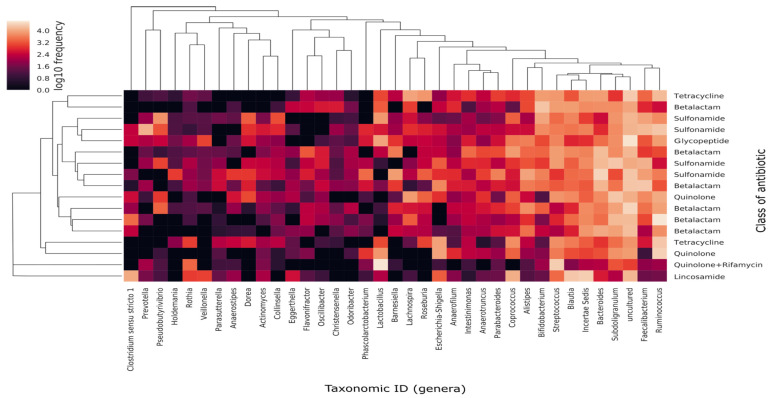
Heat map of the relative abundance and distribution of the 50 most abundant genera among samples. Each dot represents the overall bacterial community in each sample Y axis shows individual samples, tagged according to class of antibiotic and displayed following a dendrogram (left side of the graph) constructed by clustering analysis of Bray-Curtis distances among samples. X axis represents the 50 most abundant genera in terms of median of relative frequencies. Genera dendogram was constructed following Unweighted Pair Group Method using Arithmetic averages (UPGMA) clustering analysis, which groups genera according to their co-occurrence among samples.

**Figure 7 genes-12-00041-f007:**
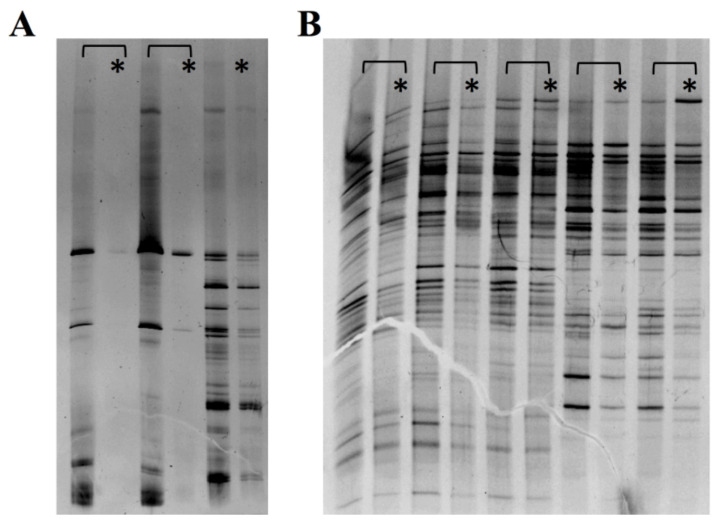
PCR-DGGE experiments for bacterial viability with (*) and without PMA. (**A**): Controls for alive bacteria respect to the same preparation but boiled to destroy bacterial cells. (**B**): Fecal samples from our patients with (*) and without PMA. The similar intensity of the bands across the experiments with and without PMA suggests that the bacteria were viable.

**Table 1 genes-12-00041-t001:** Epidemiological and clinical data of the 17 patients with SAT.

Patient	Sex (Age in Years)	Charlson Index	Location of Infection	Microorganism	Antibiotic (Family) [Months of Antibiotic Intake]	Symptomatology Attributable to SAT
1	male (79)	1	Abdominal (mesh)	MSSA	Cefadroxil (betalactam) (17)	Flatulence
2	male (73)	1	PJI (Knee)	MSSA	Doxycycline (glycopeptide) (10)	Urinary infection
3	female (83)	0	PJI (Hip)	*Enterobacter cloacae*	Ciprofloxacin (fluorquinolone) (17)	
4	female (83)	0	PJI (Hip)	MSSA	Cephalexin (betalactam) (18)	Asymptomatic bacteriuria
5	male (89)	1	PJI (Hip)	MSSA	Cephalexin (betalactam) (14)	
6	female (79)	0	PJI (Knee)	MRSE	Cotrimoxazole (109)	Diarrhoea and Asymptomatic bacteriuria
7	female (80)	1	PJI (Knee)	MSSA	Levofloxacin (fluorquinolone) /Rifamycin (1)	
8	female (88)	0	PJI (Knee)	*Escherichia coli*	Cotrimoxazole (62)	Asymptomatic bacteriuria
9	male (73)	1	Vertebral Instrumentation	Unidentified	Cotrimoxazole (54)	Constipation
10	male (84)	3	PJI (Hip)	MSSA	Cephalexin (betalactam) (19)	
11	female (70)	1	PJI (Shoulder)	*Enterococcus faecalis*	Amoxicillin (betalactam) (11)	Weight gain and Candidiasis
12	male (66)	2	PJI (Hip)	MSSE	Doxycycline (tetracycline) (48)	Diarrhoea
13	female (82)	1	PJI (Shoulder)	*Cutibacterium granulosum*	Moxifloxacin (fluorquinolone) (20)	Intestinal distension
14	female (74)	0	PJI (Knee)	MSSA	Clindamycin (macrolide) (21)	
15	male (72)	6	Osteosynthesis material (Humerus)	MRSE	Teicoplanin (glycopeptide) (1)	
16	male (34)	1	Vertebral Instrumentation	Polymicrobial	Cotrimoxazole (102)	
17	male (81)	2	Osteosynthesis material (Tibia)	MSSA	Cefadroxil (1)	

MSSA: methicillin-susceptible *S. aureus*; MRSA: methicillin-resistant *S. aureus*; MSSE: methicillin-susceptible *S. epidermidis;* MRSE: methicillin-resistance *S. epidermidis;* PJI: Prosthetic join infection. Charlson index estimated the 10-years survival: 0 = 98%, 1 = 96%, 2 = 90%, 3 = 77%, 4 = 53%, 5 = 21%, and 6 = 2%.

## Data Availability

Metagenomic sequencing datasets generated and analyzed during the current study are available in the European Nucleotide Archive under accession number: https://www.ncbi.nlm.nih.gov/bioproject/PRJNA633161.

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
