# Peer review of "Long-Term Impact of Suppressive Antibiotic Therapy on Intestinal Microbiota"

_genes, 2020, doi:10.3390/genes12010041_

Round 1

Reviewer 1 Report

The manuscript is well written and report some important issue regarding the incidence of SAT in the intestinal microbiota. Unfortunately, in my opinion this study lack of important controls. Moreover, the number of subjects is too limited to have significant results.

  1. Although well known, the significance of the Charlson index reported in table 1 should be addressed.
  2. The number of patients is limited. Most of the subjects analyzed can be classified as elderly (more than 65 years), except for one subject that is considerable young compared to the rest of the group (34 years). Since intestinal microbiota richness and diversity progressively decrease with the age, this last subject should be excluded from the analysis.
  3. This work lacks of controls (fecal samples from the same patient before SAT treatment, fecal samples from healthy individuals of the same age).
  4. The high diversity in antibiotics treatment reflects and maybe complicated the already high variability of intestinal microbiota. Significant subgroup undergoing the same antibiotics treatment should be identified in order to return consistent information.
  5. Table one is not reporting the class of antibiotics used. Although this information could appear quite futile, this makes difficult to understand the study for readers that are not expert in the field. This is further complicated by the fact that authors include antibiotics in different groups each time, thus making the comprehension more difficult. For example, at the beginning the teicoplanin was included in the list of fluoroquinolones (at least, this is what is deducible from the comparison between the text and the table 1) and then (figure 2) other class of antibiotics are reported, that are also partially different from the one reported in figure 3, where teicoplanin was reported as separated from fluroquinolones.
  6. Figure 2. The figure panel is reporting 1a and 1 b, which are possibly are mistake. Same occurred also with figure 4.

Author Response

All authors thank the reviewers for their time and recommendations. Their opinions are very important to improve the scientific quality of our manuscript.

All authors thank the reviewers for their time and recommendations. Their opinions are very important to improve the scientific quality of our manuscript.

We are aware that the lack of controls is the main problem of this work. We considered the inclusion of relatives or housemates with the same diet and environmental exposure, but it was not possible in most of our patients and finally we discarded it. Recruitment in this work has not been easy due to the particularities of the subjects who receive SAT. Despite the low number of participants, the relevance of the results seemed to us sufficient to be reported to the scientific community.

1- A paragraph has been introduced in the new version of the manuscript in relation to Charslon's index.

2- We agree with the reviewer that the difference on the patient’s age is marked, but we really consider that the greatest relevant event causing the decrease in diversity is the continuous exposition to antibiotics, much more than age. We therefore propose keeping the patient young, and not reducing the number of patients further. The age difference of the subjects has been indicated in the discussion section.

3- Indeed, the best control for this work would have been to have the stool sample from before starting the SAT, but this could not be collected as treatment had begun earlier. The main objective of this work was to analyse the safety of EAT, its impact on the selection of multi-resistant bacteria and on intestinal ecology. The pre-treatment control samples would have been valid for analysing the impact on intestinal ecology, but anyway parts of our objectives have been achieved.

4- The results are interesting but must be completed with follow-up of new patients starting with SAT, in which we will also collect adequate controls. All these issues will be considered in the future for the new work of our team in this line of research.

5- We apologize to the reviewer, the antibiotics in Table 1 have been adequately modified.

6- These errors have been corrected in the new version.

Reviewer 2 Report

The manuscript "Long-Term Impact of Suppressive Antibiotic Therapy on Intestinal Microbiota" by Escudero-Sanchez et al. investigates the diversity of intestinal microbiota in 17 patients undergoing suppressive antibiotic therapy. The manuscript is clear, well-written, and the results support conclusions presented in it. Below is the list of several minor comments:

(1) line 40: In the abstract, the authors state that they demonstrated overall safety of suppressive antibiotic therapy, which is an exaggeration. They only looked at diversity and antibiotic resistance of intestinal microbiota.

(2) line 85: In the Materials and Methods, authors specify which antibiotic concentrations they used to determine antibiotic resistance but do not specify how they chose the concentrations. Moreover, they did not test whether some bacteria could survive lower antibiotic concentrations than those they used. This fact should be pointed out explicitly in the Discussion: the SAT application might make intestinal microbiota more resistant to lower concentrations of antibiotics, which would not be detected in this study.

(3) Figure 3: It is hard to distinguish which shade of grey corresponds to which group; colour scale from Figure 2 would make Figure 3 much clearer and allow for easier comparison with results presented in Figure 2.

(4) Figure 4: Panels A and B are marked as 3a and 3b. Please remedy.

(5) Figure 5, lower panels: Description of the results does not correspond to what the Figure shows. Specifically, lines 203-206 specify taxa which are not indicated in the figure panel (Peptostreptococcaceae, Psychrobacter...)

(6) line 220: The authors specify that data from patients treated with beta-lactams cluster together, yet dendrogram on the left of Figure 6 shows this is not the case; three beta-lactam samples cluster together and three do not.

(7) line 233: The authors state that "almost all faecal bacteria present in faeces were alive and viable," but this is not what was tested. The same results would be obtained if only 10% of the present bacteria were alive, as long as the ratios of alive cells among different taxa remained somewhat unchanged.

Author Response

All authors thank the reviewers for their time and recommendations. Their opinions are very important to improve the scientific quality of our manuscript.

All authors thank the reviewers for their time and recommendations. Their opinions are very important to improve the scientific quality of our manuscript.

1- SAT safety was our main objective, and for its evaluation, a clinical follow-up has been carried out during the time indicated for each patient. The symptomatology detected for each patient is reflected in Table 1, and we reported that SAT is a safe treatment since no relevant secondary events have occurred, in addition to the low selection of multi-resistant bacteria to antibiotics. We have re-phrase, but in our experience SAT is a clinically safe treatment.

2- The concentrations used are those habitually used to test for the presence of resistant bacteria in our institution. Obviously we could have used lower concentrations and perhaps found strains that tolerated them, but these concentrations are based on previous reports in the literature and are the ones that all clinical laboratories use in epidemiological slides to look for resistant strains (with habitual molecular mechanisms responsible of these resistances).

3- We agree with the reviewer, and figure 3 has been modified to improve its understanding.

4- It has been corrected in the new version.

5- We would like to apologise for the error, which has already been corrected in the new version.

6- We agree with the observation, and it has been corrected in the new version.

7- The statement that almost all the bacteria were alive is based on the intensity of the bands obtained in the PCR-DGGE, in the case of a disproportion between live and dead bacteria a decrease in the intensity of the bands would have been observed. However we agree with the reviewer, and the sentence has been softened in the new version.

Round 2

Reviewer 1 Report

Authors have fully replied to my comments.